# Fischer–Tropsch Synthesis Catalysts for Selective Production of Diesel Fraction

Kristina Mazurova [1], Albina Miyassarova [1], Oleg Eliseev [2,3], Valentine Stytsenko [1], Aleksandr Glotov [1] and Anna Stavitskaya [1,*]

1   Department of Physical and Colloid Chemistry, Gubkin University, 65 Lelinsky Prospect,
    Moscow 119991, Russia; mazurovachris55@mail.ru (K.M.); miassarowa.albina@yandex.ru (A.M.);
    vds41@mail.ru (V.S.); glotov.a@gubkin.ru (A.G.)
2   N.D. Zelinsky Institute of Organic Chemistry, Russian Academy of Sciences (RAS), 47 Lelinsky Prospect,
    Moscow 119991, Russia; oleg@server.ioc.ac.ru
3   Institute of Organic Chemistry, Peoples' Friendship University of Russia (RUDN University),
    6 Miklukho-Maklaya St., Moscow 117198, Russia
*   Correspondence: stavitsko@mail.ru

**Abstract:** The Fischer–Tropsch process is considered one of the most promising eco-friendly routes for obtaining synthetic motor fuels. Fischer–Tropsch synthesis is a heterogeneous catalytic process in which a synthesis gas ($CO/H_2$) transforms into a mixture of aliphatic hydrocarbons, mainly linear alkanes. Recently, an important direction has been to increase the selectivity of the process for the diesel fraction. Diesel fuel synthesized via the Fischer–Tropsch method has a number of advantages over conventional fuel, including the high cetane number, the low content of aromatic, and the practically absent sulfur and nitrogen impurities. One of the possible ways to obtain a high yield of diesel fuel via the Fischer–Tropsch process is the development of selective catalysts. In this review, the latest achievements in the field of production of diesel via Fischer–Tropsch synthesis using catalysts are reviewed for the first time. Catalytic systems based on $Al_2O_3$ and mesoporous silicates, such as MCM-41, SBA-15, and micro- and mesoporous zeolites, are observed. Together with catalytic systems, the main factors that influence diesel fuel selectivity such as temperature, pressure, $CO:H_2$ ratio, active metal particle size, and carrier pore size are highlighted. The motivation behind this work is due to the increasing need for alternative processes in diesel fuel production with a low sulfur content and better exploitation characteristics.

**Keywords:** diesel fuel; Fischer–Tropsch; bifunctional catalysts; process factors





## 1. Introduction

More than eighty percent of the world's energy consumption is met by crude oil [1]. Due to the depletion of petroleum resources, the search for alternative fuel production technologies is of great demand. The Fischer–Tropsch process opens up the possibility for the production of fuels and chemicals from biomass, natural gas, or other resources and plays an increasingly important role in the energy sector.

Fischer–Tropsch synthesis is a catalytic process that could be tuned to meet various needs. The main products of this process include a wide range ($C_1$–$C_{70+}$) of hydrocarbons, primarily *n*-alkanes and linear olefins. FTS products also include iso-alkanes and cyclic hydrocarbons. Oxygenated species, such as aldehydes, ketones, acids, and alcohols, are formed during reactions under specific conditions together with $CO_2$ [2]. Technologies such as gas to liquid (GTL) and the Fischer–Tropsch synthesis to olefins (FTO) are now the most industrially demanded due to perspectives about their use as ecologically friendly alternatives to traditional production methods [3,4].

Gas-to-liquid (GTL) processes produce high-quality environmentally friendly fuels, in particular, diesel, which in terms of its performance, fully complies with Euro-5 requirements [5]. It is known that synthetic products have better characteristics compared

with refined petroleum products. Restrictions on the concentration of sulfur and aromatic hydrocarbons make diesel produced via the Fischer–Tropsch process even more interesting. In addition, synthetic diesel fuel is characterized by a cetane number of about 70, which is much higher compared with the same parameter of fuel derived from an oil refinery (about 50), a low boiling point of 90% of fuel, low density, and better biodegradability [6].

For the selective production of diesel fuels in industry, third-generation technology developed by Shell is used. A distinctive feature from the first two generations is associated with a two-stage method for obtaining motor components. The first stage is to obtain solid hydrocarbons with maximum selectivity [7]. At the second stage, after separation from liquid hydrocarbons, the paraffin fraction is sent to hydrocracking and hydroisomerization to obtain high-quality fuel. However, the multi-stage process, as a rule, leads to large energy, capital, and economic costs. In addition, the presence of hydrogen plants for the corresponding hydroprocesses on offshore facilities can create a number of safety problems [8].

Since the beginning of the 21st century, the one-stage Fischer–Tropsch process using a bifunctional catalyst has become the most attractive method for the direct conversion of synthesis gas into middle distillates with high selectivity [9]. Many scientific groups from different countries are actively working on the implementation of the fourth generation of catalysts that combine the functions of chain growth and selective hydrogenolysis.

Despite the long history of the Fischer–Tropsch process, some key issues still remain uncertain. One of the most important but difficult tasks in the production of liquid hydrocarbons is the control of selectivity [10]. Catalysts play one of the most important roles in this regard. In recent work, Fischer–Tropsch catalytic system synthesis procedures and issues such as the influence of active metals, promoters, and supports on activity and selectivity were reviewed [11,12]. However, these are mainly devoted to the selective production of a wide fraction of $C_{5+}$ hydrocarbons. The discussion of the parameters that influence the selectivity for diesel fraction is a focus of the present work. Factors affecting the yield of $C_{10}$–$C_{20}$ hydrocarbons will be discussed, and the most significant studies aimed at increasing the selectivity for diesel will be reviewed.

The used of FTS as a technology for diesel fuel production may open new possibilities for the diversification of energy resources. Controlling the parameters of the process as well as development of new catalytic systems may result in diesel fuel of very high quality. These technologies may be especially interesting during a period of elevated global oil prices [4].

## 2. The Main Factors Affecting the Selectivity of Diesel Fuels

In general, Fischer–Tropsch synthesis products follow an Anderson–Schulz–Flory (ASF) distribution due to a polymerization mechanism, according to which CO undergoes dissociative adsorption in the presence of hydrogen on the surface of active metal phases (ruthenium, cobalt, or iron) with the formation of methine, methylene, and methyl groups. The CHx groups are linked together into alkyl chains, which leads to the formation of intermediate products with different numbers of carbon atoms, which are then subjected to hydrogenation and dehydrogenation to give alkanes and olefins [13]. The hydrocarbon products of the Fischer–Tropsch process are generally divided into gases ($C_1$–$C_4$), gasoline ($C_5$–$C_{11}$), diesel distillates ($C_{12}$–$C_{20}$), and heavy paraffins ($>C_{20}$) [14]. ASF distribution is rather broad (the polydispersity index $M_W/M_N = (1 + \alpha)$, where $\alpha$ is a chain growth index) and indiscriminate with respect to fuel fractions. Thus, the highest selectivity for hydrocarbons of the gasoline and diesel series is 48% ($\alpha = 0.76$) and 30% ($\alpha = 0.88$), respectively (Figure 1).

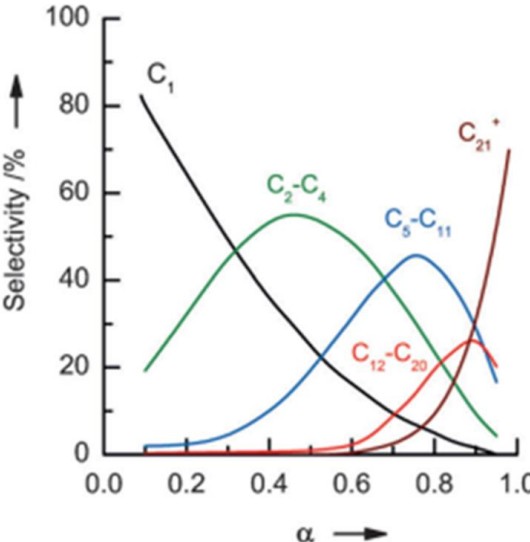

**Figure 1.** Selectivity of the Fischer–Tropsch synthesis products from the chain growth index ($\alpha$). Reprinted with permission from ref. [15]. Copyright 2013 John Wiley and Sons.

Many studies are devoted to the study of factors affecting catalytic behavior, including product selectivity. It has been established that activity and selectivity depend not only on the design of the reactor and operating conditions [16] but also on the metal used, its chemical state and crystalline phase, support, promoter, particle size, and location of the active metal [17–19]. Here, we highlight some recent research that includes basic concepts or strategies for the preparation of FT catalysts with enhanced diesel content.

*2.1. Fischer–Tropsch Process Conditions*

Process conditions significantly affect the yield of liquid hydrocarbons. Pressure and temperature change the process of primary olefin reabsorption on catalytic sites and affect selectivity for methane. A characteristic of all FT catalysts is that, with increasing temperature, the selectivity of the process shifts towards products with a lower number of carbon atoms. Under Fischer–Tropsch conditions, the initially formed $\alpha$-olefins will enter secondary reactions and participate in chain initiation, which has a positive effect on the chain growth index. With an increase in temperature from 170 °C to 215 °C, the rate of olefin readsorption increases, but a further increase in temperature kinetically increases chain termination. Nevertheless, with a further increase in temperature, the role of readsorption of $\alpha$-olefins weakens due to the relatively low concentration of olefins in the products [20]. An increase in the total pressure has a positive effect on the reaction rate and selectivity of the FTS [21]. Increasing the total pressure generally shifts product selectivity towards heavier products. With an increase in pressure, the ratio of H to CO atoms on the active surface decreases, which leads to the suppression of methane formation [22]. Also, an increase in pressure reduces the surface velocity of the gas and increases the partial pressure of olefins, which leads to increased readsorption of olefins. An increase in the $H_2$/CO ratio (from 0.5 to 2.0) leads to the production of lighter hydrocarbons and lower olefins [23]. A decrease in a space velocity of the synthesis gas leads to an increase in CO conversion due to an increase in the contact time of the feed molecules with catalytic sites. Therefore, an increase in the ratio to $H_2$/CO and a lower space velocity are favorable for the suppression of methane generation and a higher chain growth capability. Under typical Fischer–Tropsch conditions (t = 210–240 °C, 2 MPa, $H_2$/CO = 2), $C_{5+}$ selectivity increases and the olefin-to-paraffin ratio and methane selectivity decrease with increasing CO conversion. An increase in chain growth and a decrease in the ratio of olefins to paraffins are explained by an enhanced secondary reaction (resorption and re-initiation) of $\alpha$-olefins with a long residence time in the catalyst layer [24,25]. Osa et al. [26] studied the influence of the key characteristics of $Al_2O_3$-based Co- and Ca/Co Fischer–Tropsch catalysts for the production

of diesel fuel. It has been shown that CO conversion is highly dependent on the reaction temperature increase. As expected, an increase in the reaction temperature led to a gradual increase in the share of $C_1$–$C_4$ (light hydrocarbons) and CO conversion, while the share of diesel fractions decreased (Table 1). An increase in space velocity from 4000 to 12,000 $h^{-1}$ led to a decrease in CO conversion, which adversely affects chain growth [26]. In addition, it was found that the greatest readsorption of olefins and their further introduction into the hydrocarbon chain with the predominant formation of the diesel fraction is observed at a space velocity of 6000 $h^{-1}$ and 12,000 $h^{-1}$. Along with this, the effect of the $H_2$/CO molar ratio from 0.5 to 2.0 was also studied. It has been reported that a higher $H_2$/CO ratio is preferable for chain termination, leading to the formation of light hydrocarbons, while a lower ratio is necessary for chain growth, i.e., production of liquid hydrocarbons. This may be due to the fact that a high partial pressure of hydrogen leads to enrichment of the catalyst surface with hydrogen particles, which prevents the association of carbon particles into longer chains due to readsorption and hydrogenation of olefins with the formation of light hydrocarbons [27,28]. The study concluded that the promoted catalyst retained significant liquid selectivity over a wide range of $H_2$/CO molar ratios, although diesel fuel composition and CO conversion improved at an $H_2$/CO molar ratio of 2.0. The overall pressure of the process also has a significant effect on the yield of diesel fuels. In [29], at pressures of 0.4, 2, and 4 MPa, cobalt catalysts based on niobium oxide that were not promoted and promoted with alkaline earth metals were studied. It was shown that the selectivity for diesel fraction and $C_{5+}$ hydrocarbons reaches a maximum at a reaction pressure of 2.0 MPa, and the prepared catalysts had good stability. Bianchi et al. [30] also observed an increase in the diesel fraction with an increase in pressure from 0.5 MPa to 2.0 MPa during CO hydrogenation on a $Co/SiO_2$ catalyst.

**Table 1.** Catalytic characteristics of systems depending on the conditions of the Fischer–Tropsch synthesis *.

| Catalyst | Loading, wt.% | Reaction Conditions | | | | Catalysis Performance | | |
|---|---|---|---|---|---|---|---|---|
| | | T, °C | P, MPa | GHSV, $h^{-1}$ | $H_2$/CO Ratio | Conversion CO, % | $C_{5+}$ Selectivity, % | $C_{10}$–$C_{20}$ Selectivity, % |
| $Co/Al_2O_3$ | 20Co | 220 235 300 | 2 | 6000 | 2 | 25.6 32.1 86.2 | 95.50 91.64 87.44 | 70.27 65.55 65.38 |
| $Ca$-$Co/Al_2O_3$ | 0.6Ca 17Co | 220 242 | 2 | 6000 | 2 | 33.0 100 | 96.67 39.37 | 68.07 64.80 |
| $Co/Al_2O_3$ | 20Co | 220 | 2 | 4000 6000 | 2 | 34.7 25.6 | 95.76 95.50 | 39.01 52.78 |
| $Ca$-$Co/Al_2O_3$ | 0.6Ca 17Co | 220 | 2 | 4000 6000 12,000 | 2 | 55.7 33.0 6.3 | 97.91 96.67 88.46 | 55.51 67.83 64.77 |
| $Co/Al_2O_3$ | 20Co | 220 | 2 | 6000 | 0.5 1.0 1.5 2.0 | 6.6 16.6 11.0 25.6 | 96.32 97.90 90.30 95.50 | 33.98 62.68 56.57 70.66 |
| $Ca$-$Co/Al_2O_3$ | 0.6Ca 17Co | 220 | 2 | 6000 | 0.5 1.0 1.5 2.0 | 7.1 12.2 12.5 33.0 | 94.63 96.20 91.82 96.67 | 58.57 50.52 58.91 67.19 |

* The Fischer–Tropsch process was carried out on a bench-scale Inconel fixed bed reactor [26].

### 2.2. Active Phase Particle Size and Chrmical State

The most preferred active phases for Fischer–Tropsch catalysts are the Fe, Co, and Ru metals [31]. Iron-based catalysts have a high product selectivity towards lower olefins. Therefore, Fe catalysts are widely used in the study of the Fischer–Tropsch reaction for olefins. Co-based catalysts have a good ability to form long chain hydrocarbons in reaction products [32]. Ru-based catalysts have the highest activity compared with Fe and Co, high selectivity towards long chain hydrocarbons, and stability under severe reaction

conditions [33]. Therefore, from the point of view of obtaining diesel fuel, it is important to consider catalysts based on Ru and Co.

The chemical state of the active phase is critical in the development of active and selective Fischer–Tropsch catalysts. The active phases for catalysts based on ruthenium and cobalt are nanoparticles of metallic ruthenium and cobalt ($Ru^0$ or $Co^0$) [24–37]. Cobalt metal can exist in two crystalline phases: a hexagonal close-packed (hcp) phase and a face-centered cubic (fcc) phase. At the same time, hcp-Co is characterized by higher activity compared with fcc-Co. This difference is due to the fact that the hexagonal close-packed phase has a denser active center and requires less activation energy for CO adsorption (Figure 2) [38]. In addition, the structure of hcp-Co facilitates the incorporation of surface carbide for further growth of the hydrocarbon chain, which leads to the formation of heavier components [39].

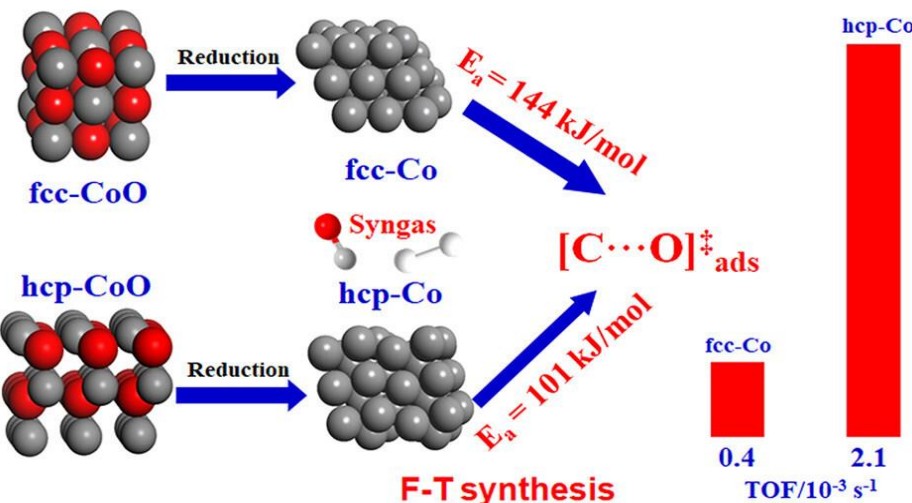

**Figure 2.** Single-phase unsupported Co catalysts for understanding the key elementary step of the Fischer–Tropsch synthesis. Reprinted with permission from Ref. [38]. Copyright 2018 American Chemical Society.

Another factor that determines catalyst behavior in the Fischer–Tropsch process is the particle size of the active phase. This parameter affects the selectivity for liquid fuels [40–48]. Reducing the average particle size of cobalt (from 27 nm to about 6 nm) leads to an increase in the dispersion of metal particles and, consequently, an increase in the activity of the Fischer–Tropsch catalyst from 0.64 to 3.51 $\times$ $10^{-5}$ $mol_{CO} \cdot g_{Co}^{-1} \cdot s^{-1}$ at 220 °C and 1 bar [40]. However, the activity will begin to decrease (to 0.80 $\times$ $10^{-5}$ $mol_{CO} \cdot g_{Co}^{-1} \cdot s^{-1}$) when a certain particle size of Co (less than 6–8 nm, depending on the catalyst and reaction conditions) is reached, which refers to the effect of particle size on the activity of Fischer–Tropsch catalysts [40,49]. For cobalt-based catalysts, a correlation between particle size and activity is observed. With an increase in the size of nanoparticles up to certain values (6–10 nm, depending on the catalyst and reaction conditions), the turnover frequency (TOF) for CO conversion and selectivity for $C_{5+}$ increase [40,42,44,46,48]. After that value the catalytic efficiency changes insignificantly.

In [48], Ru nanoparticles deposited on CNTs with an average size of 2.3 to 9.2 nm were obtained by impregnating CNTs with an aqueous solution of $RuCl_3$ followed by various treatments. Ru particles of 2.3–3.1 nm could be obtained via the direct reduction of absorbed ruthenium chloride. To increase the average ruthenium particle size to 4.0 nm, a calcination step should be introduced prior to the reduction procedure. The CO conversion rate increases ($\sim$0.05 $s^{-1}$ to $\sim$0.2 $s^{-1}$) with an increase in the average particle size from 2.3 to 6.3 nm. The CO conversion rate as well as the selectivity for liquid fuels change insignificantly after the average particles size of Ru reaches 6.3 nm. With increase in contact time to 1.5 $s \cdot g_{cat}$ $mL^{-1}$ the $C_{10}$–$C_{20}$ selectivity increased while the $C_{21+}$ selectivity decreased due to hydrocracking.

The phenomenon of decreasing TOF with particle size below the critical point was observed in systems such as Co/ITQ-2 [49], Co/SiO$_2$ [50], Co/CS (CS = carbon sphere) [44], and Co/$\gamma$-Al$_2$O$_3$ [51]. This was explained by the easier oxidation of small particles of Co$^0$ water formed under the reaction conditions [52]. According to many studies, small cobalt particles lose activity due to the combined effect of sintering and cobalt carbide formation [53].

Particle sintering leads to various discrepancies in investigations of the effect of Co particle size on product selectivity. To solve this problem, a number of encapsulated catalysts have been developed. A team of authors showed [54] that the sintering of cobalt catalyst particles based on titanium oxide can be delayed by coating them with a thin layer of silica. In [55], Co particles with a size of about 7 nm were formed inside carbon nanotubes, and the resulting catalysts retained their stability for more than 130 h at a reaction temperature of 240 °C. In [56], the Co$_3$O$_4$ nanocrystals with different sizes depending on the reaction time were synthesized via the hydrothermal method in the presence of tetradecyltrimethlammonium bromide (TTAB). The silica-TTAB layer was added through an electrostatic interaction between the cationic (TTAB) and anionic (tetraethyl orthosilicate (TEOS)) species. To remove TTAB, calcination was carried out in air. The resulting catalyst was denoted as Cat-xh (x = 4, 8, 12), where 'xh' was the duration of hydrothermal treatment. Cheng et al. found that uniform cobalt particles ranging from 7.2 to 11.4 nm, embedded in mesoporous silica substrates, prevent the release of reaction intermediates due to the limited structure and stimulate chain growth via the carbide mechanism, resulting in catalysts with high selectivity for heavier hydrocarbons. Thus, the yield of diesel and gasoline fractions reaches 66.2% and 62.4%, respectively. Figure 3 shows the size distributions of the cobalt crystallites for the fresh catalysts and the values of the selectivity of the process depending on the size of the crystallites.

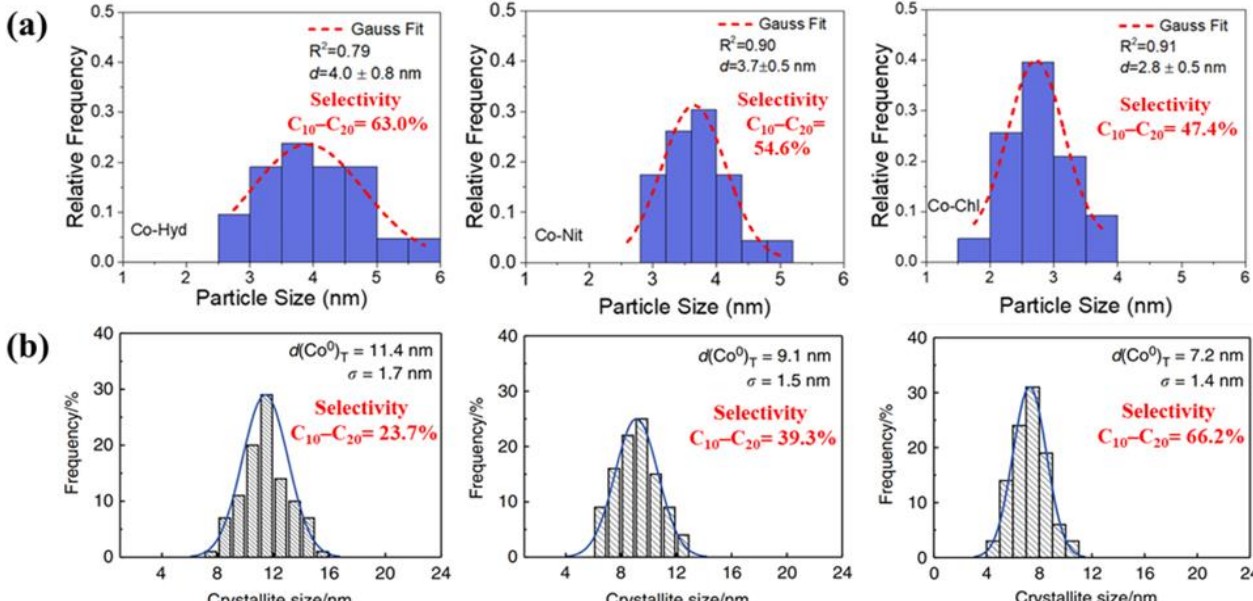

**Figure 3.** Crystallite size distributions of the reduced catalysts published in [53] (**a**) and [56] (**b**). Reprinted with permission from Ref. [53] (copyright 2019 Elsevier) and Ref. [56] (copyright 2018 Springer Nature).

The authors of [39,57] also studied cobalt catalysts with different particle sizes and different crystallographic phases of Co (hcp-Co and fcc-Co). As a result, it was shown that systems with a smaller particle diameter of cobalt (2.4 nm) in the presence of a hexagonal close-packed crystalline phase provide a higher rate of CO dissociation and selectivity for C$_{5+}$ hydrocarbons compared with fcc-Co. Table 2 shows a summary of the effect of particle size on the selectivity of the Fischer–Tropsch process.

**Table 2.** Catalytic efficiency of Fischer–Tropsch process systems as a function of active phase particle size.

| Catalyst | Loading, wt.% | Particle Size [a], nm | Particle Size [b], nm | Catalysis Performance | | | Ref. |
|---|---|---|---|---|---|---|---|
| | | | | Conversion CO, % | $C_{5+}$ Selectivity, % | $C_{10}$–$C_{20}$ Selectivity, % | |
| IC [c] | | 12.4 | 12.1 | 75 | 24.9 | | |
| MC1 [c] | 15Co | 7.9 | 7.6 | 89 | 26.8 | | [43] |
| MC2 [c] | | 8.6 | 8.3 | 85 | 26.9 | | |
| MC3 [c] | | 9.8 | 9.4 | 81 | 26.4 | | |
| Ru/CNT-C573-R673 [d] | | 7.4 | | 30 | | 65 | [48] |
| 30%Co/SiO$_2$ [e] | 30Co | 183 | | 10 | 78.9 | | |
| 10%Co/ITQ(1) [e] | 10Co | 12.8 | | 10 | 61.6 | | [49] |
| 10%Co/ITQ(2) [e] | 10Co | 8.2 | | 10 | 63.0 | | |
| Co/CNTs.A [f] | | 5.1 | 5 ± 0.2 | 16.4 | 18.4 | | |
| Co/CNTs.A.600 [f] | | 4.2 | 4 ± 0.2 | 28.3 | 33.2 | | |
| Co/CNTs.A.700 [f] | 10Co | 5.3 | 5 ± 0.2 | 37.5 | 38.8 | | [55] |
| Co/CNTs.A.800 [f] | | 6.1 | 6 ± 0.2 | 50.9 | 54.6 | | |
| Co/CNTs.A.900 [f] | | 7.2 | 7 ± 0.2 | 58.7 | 59.1 | | |
| Cat-12h [g] | 13.9Co | 11.4 | | 80.6 | 80.0 | 23.7 | |
| Cat-8h [g] | 14.5Co | 9.1 | | 78.2 | 74.9 | 39.3 | [56] |
| Cat-4h [g] | 15.3Co | 7.2 | | 77.0 | 84.2 | 66.2 | |
| Cat-1M [g] | 16.3Co | 14.3 | | 84.0 | 67.3 | 29.7 | |

[a] Determined from TEM. [b] Determined from XRD. [c] FT synthesis was performed in a tubular fixed-bed microreactor. FTS reaction conditions: T = 220 °C, P = 2 MPa, H$_2$/CO = 2. [d] FT synthesis was performed on a fixed-bed high-pressure stainless-steel reactor. FTS reaction conditions: T = 260 °C, P = 2 MPa, and H$_2$/CO = 1. [e] FT synthesis was carried out in a downflow fixed-bed stainless steel reactor. FTS reaction conditions: T = 220 °C, P = 2 MPa, and H$_2$/CO = 2. [f] FT synthesis was performed in a continuous flow fixed-bed reactor. FTS reaction conditions: T = 260 °C, P = 2 MPa, and H$_2$/CO = 1. [g] FTS reaction conditions: T = 220 °C, P = 2 MPa, and H$_2$/CO = 2. Note: In [48], all silica delaminated ITQ-2 zeolite was synthesized following the procedure developed at the ITQ and reported elsewhere [57]. Model Co/ITQ-2 catalysts are denoted as 10%Co/ITQ(x), where ITQ is zeolite ITQ-2 and (x) is the microemulsion used in their preparation.

### 2.3. Carriers

The nature of the support has a significant effect on the activity and selectivity of the catalyst [58,59]. Also, the role of the carrier is to increase the thermal stability of the catalyst by reducing the likelihood of sintering [60]. The substrate affects not only the diffusion of the reagents but also the reducibility and dispersity [61]. Textural characteristics of carriers, such as specific surface area, pore volume, and pore size distribution, affect the distribution of the active phase on the surface of the carrier and, consequently, the interaction of the active phase with the carrier [62]. Too strong an interaction of the support with the metal leads to the formation of hard-to-recover compounds, and a weak interaction reduces the dispersity of the catalyst [63]. In addition, the stability of the resulting catalytic systems is also affected by the main physicochemical characteristics of the support [64–69].

The most widely studied carriers of catalysts for the Fischer–Tropsch process are oxides of silicon, aluminum, and titanium [70–73]. Alumina is an attractive carrier due to its excellent mechanical properties, high abrasion resistance, and controlled pore structure [74,75]. The main drawback of aluminum oxide is associated with its strong interaction with metal particles, which leads to the formation of spinels and, hence, to a decrease in the number of active centers [76]. Compared with alumina, silica is characterized by a lower metal–carrier interaction, a large surface area, a narrow pore size distribution, and thermal stability of the catalyst [77,78]. Titanium oxide has a high specific surface area and corrosion resistance [79]. Recently, structured materials, such as zeolites of various compositions such as ZSM-5, SAPO-11, silicates, and aluminosilicates (MCM-41, Al-MCM-41, SBA-15), have become widely known in the Fischer–Tropsch synthesis process [80,81]. Unlike amorphous supports, the proposed materials can improve catalytic efficiency by easily controlling the catalyst structure (morphology), which can affect the activity, selectivity, and stability of the prepared system [82,83].

Carbon-based supports such as activated carbon (AC), carbon nanotube (CNT), carbon nanofiber (CNF), carbon sphere (CS), and metal–organic framework (MOF)-derived carbonaceous materials for FTS catalysts have been reviewed [84]. The drawbacks of

carbon-based materials such as weak mechanical strength, weak stability, and inability to regenerate in the oxidizing atmosphere can be overcome by using hybrid structures, composed of both oxide and carbon.

### 2.4. Microporous and Mesoporous Materials

The yield of diesel fractions can be enhanced by regulating carrier porosity. It is assumed that the microporous structure mainly leads to the production of gasoline fuels, while the mesoporous material has a positive effect on mass and heat transfer and promotes the growth of the hydrocarbon chain with the production of heavier fractions.

The catalytic efficiency of impregnated cobalt catalysts based on $SiO_2$; $Al_2O_3$; montmorillonite; and zeolites USY, ZSM-5, or MCM-22 was studied in the suspension phase of Fischer–Tropsch synthesis at 1.0 MPa and 230 °C [85]. When using an acid carrier, the average molecular weight of the synthesis products shifted toward gasoline hydrocarbons ($C_4$–$C_{12}$) and the share of heavy fuels decreased. According to the authors, this is due to the fact that micropores limit the growth of the hydrocarbon chain. Due to the microporous structure and acidity, the selectivity for iso-paraffins increased. The $SiO_2/Al_2O_3$ ratio also influenced the selectivity. Its increase from 30 to 80 in zeolite (reduction in aluminum content) showed an increase in activity and a decrease in methanation and selectivity for iso-paraffins.

Khodakov et al. [86] studied the dependence of the degree of reduction and dispersity of the cobalt catalyst on the porosity of the carrier. They found that the reducibility of Co compounds is higher with large pore silica (pore diameter = 33 nm) due to larger $Co_3O_4$ crystallites that are easier to reduce. For catalysts with narrower pores (7.5–9.0 nm), the formation of smaller cobalt particles was characteristic; lower CO conversion and selectivity for liquid products were observed. The carriers used in this work were SBA-15 silica (designated as S1) and the second support was obtained from a commercial fumed silica (Cab-osil M5, Cabot) (designated as S2).

Gonzalez et al. [87] studied a number of catalysts based on cobalt (20% wt.) and silica-based mesoporous molecular sieve (SBA-15, Al-MCM-41, and INT-MM1) as well as commercial amorphous silica for comparison. The activity was evaluated at 523 K, 10 bar, and $H_2/CO$ = 2. A great influence of catalyst porosity on catalyst structure, recoverability, and activity was found. Larger cobalt oxide particles formed in wide pores, which favorably affected their reducibility and activity. In particular, selectivity by diesel fraction increased. The best result was shown in the Co/SBA-15 catalyst. CO conversion was equal to 40%, and $C_{5+}$ selectivity was equal to 80%. The Al-MCM-41 and INT-MM1-based catalysts showed lower activity due to their smaller pore size and cobalt oxide particles, which are difficult to recover.

The influence of the catalyst structure on the selectivity with respect to specific products was also described in [56]. The authors synthesized metal nanoparticles inside the pores of mesoporous silica. An increase in the selectivity for heavier hydrocarbons due to the readsorption of olefins in a limited space was assumed. In addition, it was shown that the selectivity for the products can be controlled by controlling the size of Co crystals in the range from 7.2 to 11.4 nm. It was concluded that the growth of the hydrocarbon chain depends not only on the particle size but also on the structure of the catalyst.

### 2.4.1. Effect of Acidity

Acidity of the support plays an important role in selective diesel formation in FTS. In the work of D.J. Moon et al., SBA-15 modified with aluminum (Si/Al molar ratios equaled 5, 7, and 10) was used as a carrier for cobalt-based FT catalysts [88]. It was assumed that an increase in the acidity of the carrier resulted in a better cobalt–carrier interaction that affected the activity of the catalysts as well as better selectivity for $C_{11}$–$C_{18}$ hydrocarbons.

The effect of acidity on the yield of diesel fraction was shown on cobalt catalysts containing 16 wt.% of metal supported on mesoporous gamma alumina with isolated silica sites [89]. It was shown that doping with Si makes a significant contribution to the

acidity of the support. Such catalyst characteristics resulted in moderate cracking reactions that suppressed the formation of long chain waxy hydrocarbons and raised the yield of middle distillate.

In [90], the effect of complexing agents used for the preparation of ruthenium catalysts on the yield of Fischer–Tropsch synthesis products was investigated. In this study, metal nanoparticles stabilized via ligands (ethylenediaminetetraacetic acid, urea, and aceton-azine) were formed inside aluminosilicate nanotubes (halloysite). It was shown that after reduction, these systems were characterized by different total acidity that significantly affected the selectivity of products. The highest acidity was achieved when ethylenediaminetetraacetic acid was used as a ligand, and this led to a significant yield of methane. The highest yield of liquid hydrocarbons was observed for the catalyst, which was obtained using urea as a complexing agent due to low acidity of the resulting catalytic system.

A number of studies have shown that a moderate amount of medium acid sites leads to a higher yield in relation to the production of diesel fuel. At the same time, compared with amorphous catalysts, systems based on zeolites exhibit higher selectivity with respect to $C_{10}$–$C_{20}$ hydrocarbons [91]. Li et al. [92] used cobalt nanoparticles deposited on mesoporous Y-type zeolites (Ymeso) for complex controlled synthesis of liquid fuel using FT technology. Control of the porosity and acidity of the zeolites was used to adjust the type of liquid product formed. The study showed that the acid properties and porosity of Ymeso zeolites have a great influence on the molecular weight distribution of the products. Table 3 shows the results of the Fischer–Tropsch process for catalysts with different acidities.

In [93], the Fischer–Tropsch synthesis reaction (FTS) was studied on Ru-, Pt-, and La-promoted $Co-Al_2O_3/ZSM-5$ hybrid catalysts prepared via the suspension co-precipitation method. The promoted catalysts show higher selectivity for diesel hydrocarbons compared with the unpromoted $Co-Al_2O_3/ZSM-5$ catalysts. The $Co-Al_2O_3-Pt/ZSM-5$ hybrid catalyst showed the least methanation and conversion of olefins as well as high selectivity for $C_{10+}$ hydrocarbons. The authors suggest that the decrease in selectivity towards lower hydrocarbons may be due to the suppressed cracking properties of heavy olefins due to the presence of fewer acid sites.

A series of catalysts based on cobalt supported on ZSM-5 having four different silicon/aluminum ratios was studied. In addition, a cobalt catalyst based on amorphous silicon oxide was prepared [94]. As a result, it was found that among the prepared catalysts based on zeolite, an increase in the selectivity for $C_{10+}$ hydrocarbons and a decrease in the content of olefins in the products of the Fischer–Tropsch synthesis were observed for the Co/ZSM-5 catalyst with a ratio of Si/Al = 250, which is associated with the suppression of excessive cracking of heavy hydrocarbons due to the moderate content of acid centers.

### 2.4.2. Effect of Pore Size

The catalytic performance of Fischer–Tropsch catalysts, especially the product selectivity depends on the diffusion of products and reactants, reactions at active sites, and secondary processes. It is believed that the pore size of the carrier can affect several factors: (1) the reducibility and dispersion of the metal, (2) the diffusion of products and reagents, and (3) the possibility of secondary reactions. All of these factors are critical to determining the catalyst behavior in the Fischer–Tropsch process. The development of mesoporous materials with a narrow pore distribution and their use as catalyst's carriers in FTS has made it possible to take a look at the effect of pore size.

**Table 3.** Catalytic efficiency of Fischer–Tropsch process * systems as a function of depending on the acidity of the carrier.

| Catalyst. | Loading, wt.% | Acidity | NH$_3$ Adsorption Quantity (mmol g$^{-1}$) | Reaction Conditions | | | Catalysis Performance | | | Ref. |
|---|---|---|---|---|---|---|---|---|---|---|
| | | | | $T$, °C | $P$, MPa | H$_2$/CO Ratio | Conversion CO, % | C$_{5+}$ Selectivity, % | C$_{10}$–C$_{20}$ Selectivity, % | |
| Co/Al-SBA-15(10) | | Strong | 6.81 | | | | 33.45 | 69.98 | 10 | |
| Co/Al-SBA-15(7) | | Medium | 7.55 | 230 | 2 | 2 | 32.06 | 76.18 | 13 | [88] |
| Co/Al-SBA-15(5) | | Weak | 5.44 | | | | 49.59 | 82.44 | 20 | |
| HTN@Ru-1 | | Strong | 0.315 | | | | 15.6 | 26.7 | 48.9 | |
| HTN@Ru-2 | 2Ru | Weak | 0.129 | 260 | 1 | 2 | 17.8 | 78.0 | 52.7 | [90] |
| HTN@Ru-3 | | Medium | 0.250 | | | | 18.8 | 67.7 | 67.1 | |
| Co/Ymeso-H | | Strong | - | | | | - | 66 | - | |
| Co/Ymicro-Na | | Weak | - | | | | - | 74 | - | |
| Co/Ymeso-K | 15Co | Weak | - | 250 | 2 | 1 | - | 76 | 58.0 | [92] |
| Co/Ymeso-Ce | | Medium | - | | | | - | 86 | 27.6 | |
| Co/Ymeso-La | | Medium | - | | | | - | 88 | 52.6 | |
| Co-Al$_2$O$_3$-Ru/ZSM-5 | 13Co 0.3Ru | Strong | 0.055 | | | | 31 | 76 | 55 | |
| Co-Al$_2$O$_3$-Pt/ZSM-5 | 13Co 0.3Pt | Medium | 0.056 | 240 | 2 | 2 | 41 | 80 | 62 | [93] |
| Co-Al$_2$O$_3$-La/ZSM-5 | 13Co 0.3La | Weak | 0.055 | | | | 21 | 60 | 46 | |
| Co/ZSM-5 (Si/Al = 15) | 20Co | Strong | 0.549 | | | | 56.2 | 43.2 | 18.1 | |
| Co/ZSM-5 (Si/Al = 25) | 20Co | Medium | 0.512 | 240 | 2 | 2 | 50.6 | 44.9 | 25.7 | [94] |
| Co/ZSM-5 (Si/Al = 140) | 20Co | Weak | 0.219 | | | | 47.3 | 52.4 | 38.8 | |
| Co/ZSM-5 (Si/Al = 250) | 20Co | Weak | 0.117 | | | | 43.1 | 60.4 | 47.6 | |

Note: The acidity of FTS catalysts was measured via the temperature-programmed desorption of ammonia (NH$_3$-TPD) in the range of 50–900 °C. The first peaks at temperature of about 50–250 °C in the NH$_3$-TPD spectra are considered as weak acid sites, the second peaks at temperatures of about 250–500 °C are regarded as middle acid sites, and the third peaks at temperatures of about 500–800 °C are strong acid sites. * FT synthesis was performed in a fixed-bed flow-type reactor.

The effect of pore size was studied using conventional amorphous $SiO_2$ or $\gamma$-$Al_2O_3$. As a result, using the example of Co/$SiO_2$ with a pore diameter of 2, 4, 6, 10, or 15 nm, it was shown that selectivity for $C_{5+}$ hydrocarbons, CO conversion, as well as the particle size of Co and their reducibility increase with increasing pore size to 10 nm [95]. A larger pore size results in larger particles, better reducibility, and hence high activity and selectivity for liquid hydrocarbons. In [96], using $\gamma$-$Al_2O_3$, the positive effect of increasing the pore size was also confirmed. It has been suggested that the higher selectivity for $C_{5+}$ hydrocarbons depends on the high degree of re-adsorption of $\alpha$-olefins in the wider pores, as well as on the larger Co particles.

Bartolini et al. [97] investigated the effect of the pore size of catalyst carrier SBA-15 (5.0, 10.7, and 14.6 nm) on the molecular weight distribution of products in the Fischer–Tropsch process. The catalysts with 30% of cobalt were prepared using impregnation method. Experiments showed that porosity has an effect on the recoverability and size of Co particles. A correlation was also found between pore size and hydrocarbon chain length of the products. Larger pores allow for the synthesis of larger and heavier molecules.

The authors of [98] showed that efficient control of hydrogenolysis using cobalt nanoparticles deposited on mesoporous zeolite Y can increase the selectivity of diesel fraction while maintaining a low methane yield. The particle sizes of cobalt and diameter of mesopores are the key factors determining the selectivity of FT synthesis. The selectivity for diesel in the synthesis reached 60%, with the selectivity for $CH_4$ of about 5% using the cobalt catalyst supported on mesoporous Y of the Na-type with average sizes of 8.4 nm Co particles and 15 nm mesopores. Table 4 shows the selectivity for $C_{5+}$ hydrocarbons and diesel fraction for Fischer–Tropsch catalysts prepared on supports with different pore sizes.

Thus, the structure and acidic properties of the carrier have a significant role in the selectivity of liquid hydrocarbons (Figure 4). Small catalyst pores with high acidity lead to the highest yield of the gasoline fraction, while for mesoporous systems with moderate acid sites, high selectivity for the diesel series components was observed.

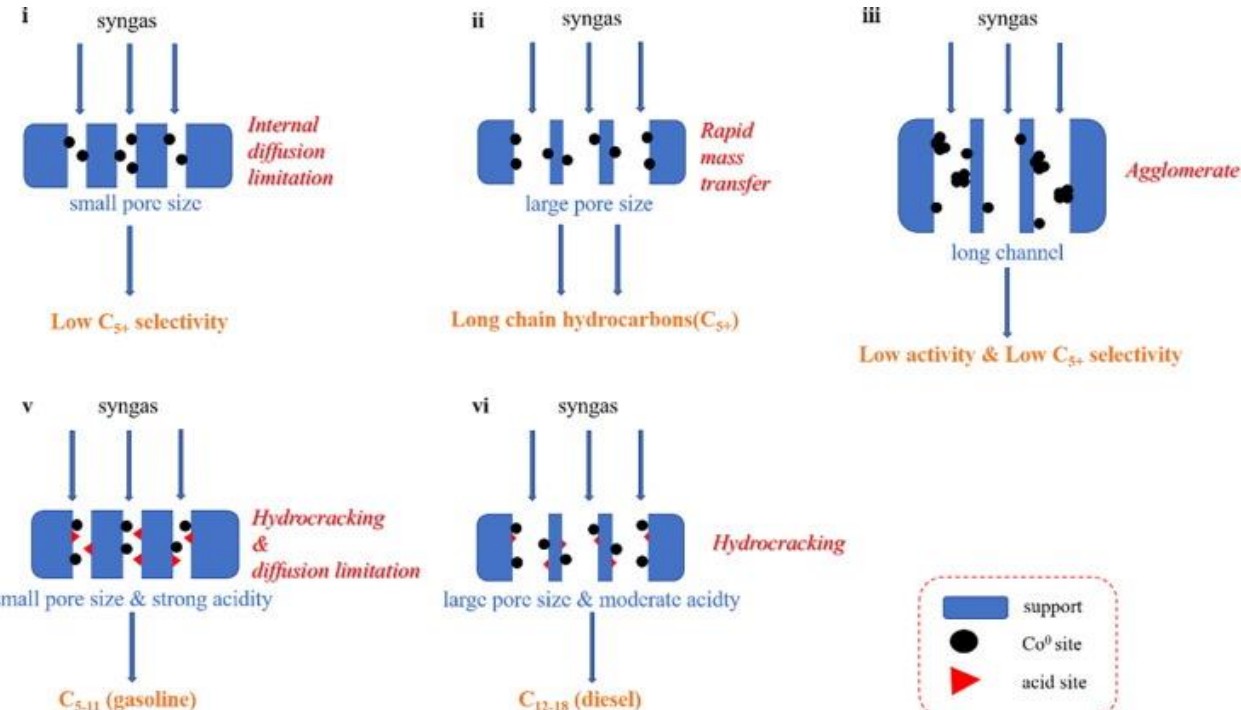

**Figure 4.** Influence of acidity and pore size of support on selectivity for liquid hydrocarbons. Reprinted with permission from ref. [11]. Copyright 2022 Elsevier.

**Table 4.** Catalytic efficiency of Fischer–Tropsch process systems as a function of depending on the pores of the carrier.

| Catalyst | Loading, wt.% | Carrier Pore Diameter, nm | Reaction Conditions | | | Catalysis Performance | | | α | Ref. |
|---|---|---|---|---|---|---|---|---|---|---|
| | | | T, °C | P, MPa | $H_2$/CO Ratio | Conversion CO, % | $C_{5+}$ Selectivity, % | $C_{10}$–$C_{20}$ Selectivity, % | | |
| Co(N)/MCM-41(IMP) | | | | | | 20 | 53 | 21 | 0.70 | |
| Co(N)/MCM-41 (TIE) | 10Co | 2.2 | 250 | 2 | 2 | 7.7 | 58 | 18 | 0.73 | [64] |
| Co(A)/MCM-41 (TIE) | | | | | | 10 | 64 | 10 | 0.70 | |
| Co(A)/SBA-15 | | | 250 | | | 14 | | 13 | 0.69 | |
| Co(A)/SBA-15 | 20Co | 3.5 | 250 | 2 | 2 | 72 | | 8 | 0.72 | [64] |
| Co(10A + 10N)/SBA-15 | | | 230 | | | 84 | | 32 | 0.93 | |
| Co(N)/SBA-15 | | | 230 | | | 89 | | 30 | 0.92 | |
| 5CoS1 | 5Co | | | | | | 68.4 | | 0.77 | |
| 10CoS1 | 10Co | 9.1 | 190 | 0.1 | 2 | 4 | 67.1 | | 0.77 | [86] |
| 50CoS1 | 50Co | | | | | | 43.1 | | 0.68 | |
| 5CoS2 | 5Co | | | | | | 60.7 | | 0.70 | |
| 10CoS2 | 10Co | 33 | 190 | 0.1 | 2 | 4 | 63.0 | | 0.74 | [86] |
| 50CoS2 | 50Co | | | | | | 63.3 | | 0.75 | |
| Co/SBA-15 | | 4.9 | | | | 63.2 | 71.1 | | 0.70 | |
| Co/Al-MCM-41 | 20Co | 3.2 | 250 | 1 | 2 | 38.5 | 74.4 | | 0.72 | [87] |
| Co/INT-MM1 | | 2.6 | | | | 40.3 | 71.8 | | 0.70 | |
| Co/SiO2 | | 4 | | | | 44 | 71 | | 0.80 | |
| Co/SiO2 | 20Co | 10 | 220 | 1.5 | 2 | 60 | 74 | | 0.85 | [95] |
| Co/SiO2 | | 15 | | | | 30 | 72.1 | | 0.84 | |
| C-2 | | 6.5 | | | | | 79.8 | | | |
| C-5 | 20Co | 9.0 | | | | | 81.3 | | | |
| C-8 | 0.5Re | 10.5 | 210 | 2 | 2 | 50 | 80.9 | | | [96] |
| C-12 | | 20.8 | | | | | 83.8 | | | |
| Co-SBA-15-1 | | 5 | | | | 28 | 58 | 36 | 0.72 | |
| Co-SBA-15-8 | 30Co | 10.7 | 230 | 2 | 2 | 29 | 54 | 41 | 0.78 | [97] |
| Co-SBA-15-11 | | 14.6 | | | | 28 | 49 | 41 | 0.83 | |

*2.5. Promoters*

Promoters play an important role in FT synthesis, especially for Co-based catalysts. Promoters are usually introduced via stepwise or co-impregnation methods. Noble-metal-promoted catalysts typically exhibit a high Co site density and consequently, higher FTS rates and $C_{5+}$ selectivity compared with unpromoted catalysts [99]. One of the most studied promoting agent for FT catalyst preparation is ruthenium [100–102]. The addition of ruthenium reduces the amount of $Co^{2+}$ and $Co^{3+}$ particles, increases the formation of active sites of cobalt (Co), and promotes easier and faster reduction of cobalt oxide.

Such an effect of ruthenium is due to the phenomenon of hydrogen spillover; its action can be described in three or four stages (Figure 5). At the first stage, the dissociative adsorption of hydrogen occurs. At the second, surface migration of H particles occurs from the promoter to the reduced oxide. At the last stage, the formation of the $Co^0$ metal phase via the interaction of atomic hydrogen with oxygen from $Co_3O_4$ occurs. This mechanism of primary hydrogen transfer occurs if the promoter and metal oxide are in close contact. If the promoter crystallite is isolated and the migration of H particles directly to the oxide is impossible, then the migration of H particles occurs in two stages (secondary hydrogen transfer), rather than one. At the first stage, the dissociative adsorption of hydrogen also occurs. At the second stage, hydrogen migrates to the carrier surface. At the third stage, surface migration of H particles to the surface of the reduced oxide occurs through surface OH groups by alternating between the breaking and formation of O-H bonds. On the fourth, the reduction of metal oxide with hydrogen occurs. There is evidence that the reduction of the oxide occurs faster during the primary hydrogen transfer than during the secondary transfer [103].

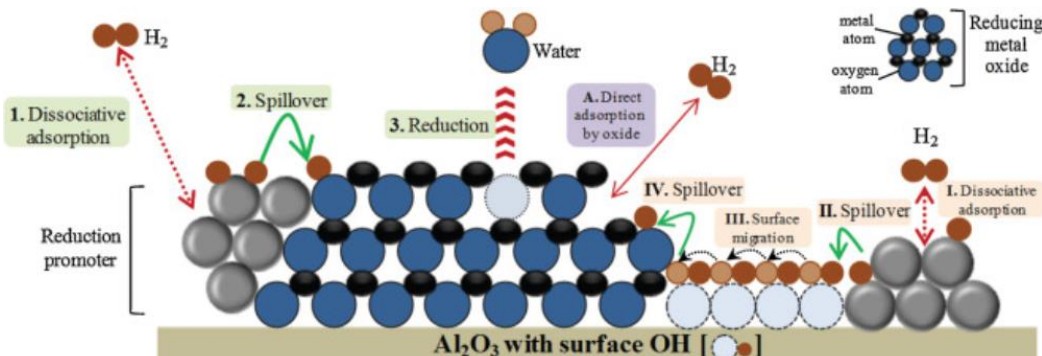

**Figure 5.** Proposed hydrogen spillover pathway in the reduction of metal oxide with the help of a promoter. Reprinted with permission from ref. [103]. Copyright 2016 Elsevier.

Hydrogen spillover directly contributes to the formation of a larger number of active metal sites and dispersion, which has a beneficial effect on the activity of the entire catalyst [104–109]. Spillover also helps to reduce catalyst deactivation by hydrogenating on its surface inactive particles of carbon or oxygen/oxides formed during the synthesis [103].

Xiong et al. [110] investigated ruthenium (0.05–0.5 wt.%) as a promoter for a cobalt catalyst supported on SBA-15. Ruthenium-promoted catalysts showed slightly higher selectivity (81.7–84.2%) for $C_{5+}$ hydrocarbons compared with the unpromoted catalyst (79.0%). An increase in the content of ruthenium led to an increase in CO adsorption by the catalysts, which improved the catalytic activity in FTS. In addition, the increased selectivity for $C_{5+}$ hydrocarbons could be associated with an increase in the electron density of Co active centers, which leads to enhanced readsorption of $\alpha$-olefins [111].

The positive effect of the noble metal on the diesel fraction selectivity was studied in [112]. The authors compared two catalysts based on cobalt: $Co/Al_2O_3$ and $Co-Re/Al_2O_3$. CO conversion and hydrocarbon selectivity were studied as a function of pressure, $CO/H_2$ ratio, temperature, and gas flow rate. The promoted cobalt catalyst after 15 days of operation allowed for a high CO conversion of 93.7%, as well as a selectivity for diesel

fraction of 62.3% to be achieved; compared with the literature data, these are very high rates. The obtained results are presumably associated with the fact that Re, having a high activity, promotes better hydrogen dissociation and the formation of CH, CH$_2$, and CH$_3$ particles, and this leads to an increase in CO conversion.

Kang et al. [113] deposited cobalt nanoparticles on mesoporous Na-Y zeolite (Na-*meso*-Y) via melt impregnation and infiltration. Compared with impregnation, melt infiltration resulted in the production of Co nanoparticles selectively within mesopores with a narrower size distribution, uniform particle distribution, high reducibility, and dispersion. It was shown that the average size of Na–*meso*-Y mesopores, as well as the size of cobalt particles, affect the selectivity of the product. The addition of manganese in the proper amount (atomic ratio Mn/Co = 0.21) can further improve diesel fraction selectivity up to 65% by suppressing the formation of CH$_4$ and lighter hydrocarbons, but at the same time, it reduces conversion. The catalyst Co/Na-*meso*-Y modified with manganese was not subjected to deactivation for 1000 h, which characterizes its stability.

In [26], the effect of reaction conditions on the Fischer–Tropsch activity and product distribution of a cobalt catalyst based on calcium-promoted alumina was studied. It was noted that the addition of a small amount of calcium oxide as a promoter (0.6 wt.%) improved the reduction of cobalt oxide and reduced the formation of difficult-to-reduce metal-supported compounds. According to the FTS results, it was found that the addition of Ca increased CO conversion and selectivity for C$_{5+}$ hydrocarbons. It is also important to note that the presence of calcium in the catalyst composition increased the yield of diesel hydrocarbons.

Table 5 presents the results of the Fischer–Tropsch process for C$_{5+}$ hydrocarbons and diesel fraction for catalysts with promoter.

**Table 5.** Promoter effect on Catalytic performance of Fischer–Tropsch process * systems.

| Catalyst | Promoter | Loading, wt.% | Reaction Conditions | | | Catalysis Performance | | | α | Ref. |
|---|---|---|---|---|---|---|---|---|---|---|
| | | | T, °C | P, MPa | H$_2$/CO Ratio | Conversion CO, % | C$_{5+}$ Selectivity, % | C$_{10}$–C$_{20}$ Selectivity, % | | |
| Co-Ru/Ti-A [a] | Ru | 10Co 0.5Ru | 220 | 2 | 2 | 10 | 62.0 | | | [100] |
| Co-Ru/Ti-R [a] | Ru | 10Co 0.5Ru | | | | 10 | 79.1 | | | |
| Co/Ru/Al$_2$O$_3$ | Ru | 15Co 0.15Ru | | | | 15.3 | 40.7 | | 0.65 | [101] |
| Ru/Co/Al$_2$O$_3$ | Ru | 0.15Ru 15Co | | | | 12.9 | 43.1 | | 0.56 | |
| CoRu/Al$_2$O$_3$ | Ru | 15Co 0.15Ru | 220 | 2 | 2 | 19 | 46.5 | | 0.72 | |
| CoRu/Al$_2$O$_3$ | Ru | 15Co 0.3Ru | | | | 21.1 | 44.2 | | 0.70 | |
| CoRu/Al$_2$O$_3$ | Ru | 15Co 0.6Ru | | | | 10.2 | 45.7 | | 0.69 | |
| CoRu1 [b] | Ru | 15Co 0.05Ru | | | | 24.7 | 42.1 | | 0.70 | [102] |
| CoRu2 | Ru | 15Co 0.1Ru | | | | 26.8 | 43.9 | | 0.73 | |
| CoRu3 | Ru | 15Co 0.15Ru | 220 | 0.1 | 2 | 31.1 | 46.4 | | 0.73 | |
| CoRu4 | Ru | 15Co 0.2Ru | | | | 40.1 | 44.3 | | 0.73 | |
| CoRu5-180 | Ru | 15Co 0.3Ru | | | | 42.9 | 44.9 | | 0.74 | |
| Co/Ru/SBA-15 | Ru | 30Co 0.05Ru | 210 | 2 | 2 | 33.4 | 81.7 | | | [110] |
| Co/Ru/SBA-15 | Ru | 30Co 0.1Ru | | | | 35.4 | 84.2 | | | |
| Co-Re/Al$_2$O$_3$ | Re | 20Co 0.5Re | 230 | 0.85 | 2.3 | 94 | 89 | 62 | | [112] |
| Co-Ru/Al$_2$O$_3$ | Ru | 15Co 1Ru | 240 | 240 | 2 | 2 | 50 | 69 | | |
| Co-Rh/Nb$_2$O$_5$ | Rh | 1.9Co 2.3Rh | 150 | 0.1 | 2 | 26 | 68 | n.a. | | |
| Co-Ni-Cs/La$_2$O$_3$ | Ni, Cs | 50Co 20Ni1Cs | 260 | 0.2 | 2 | 20 | 21 | n.a. | | |
| Co-Ru/Na-*meso*-Y-IMF | Ru | 15Co 0.3Ru | | | | 45 | 85 | 52 | | [113] |
| Co-Re/Na-*meso*-Y-IMF | Re | 15Co 0.3Re | | | | 39 | 88 | 54 | | |
| Co-Ce/Na-*meso*-Y-IMF | Ce | 15Co 3Ce | 230 | 2 | 1 | 21 | 75 | 38 | | |
| Co-Zr/Na-*meso*-Y-IMF | Zr | 15Co 3Zr | | | | 32 | 86 | 49 | | |
| Co-Mn/Na-*meso*-Y-IMF | Mn | 15Co 3Mn | | | | 37 | 92 | 65 | | |

* Fischer–Tropsch synthesis was performed in a down-flow fixed bed stainless steel reactor. [a] The TiO$_2$ anatase and rutile supports are labeled as Ti-A and Ti-R, respectively. [b] For this series of catalysts, the carrier was γ-Al$_2$O$_3$.

## 3. Bifunctional Catalysts

The yield of liquid hydrocarbons and feedstock conversion can be adjusted by changing the chemical state and the crystalline phase of the active metal; adjusting the particle and pore size of the support; as well as choosing the appropriate promoters and, of course, the Fischer–Tropsch synthesis conditions. However, new strategies are required to obtain higher selectivity for middle distillate hydrocarbons such as diesel fuel.

Bifunctional catalysts combine a traditional FT catalyst for the hydrogenation of CO to heavier hydrocarbons and an acid site on which hydrocarbons undergo hydrocracking. And, recently, such catalysts have been considered for the process of direct conversion of synthesis gas with the production of middle distillates with high selectivity [15,92,93,98,113–115] (Figure 6).

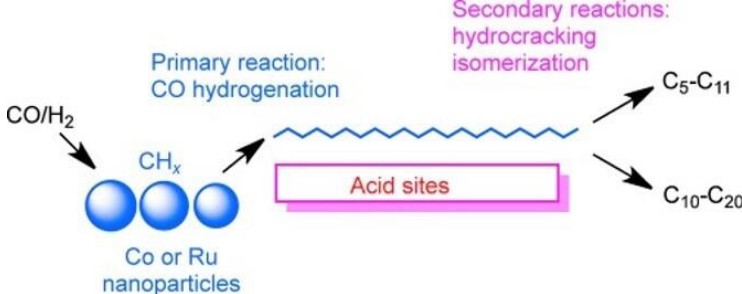

**Figure 6.** Scheme for obtaining liquid hydrocarbon fractions on bifunctional catalysts of the Fischer–Tropsch process. Reprinted with permission from ref. [15]. Copyright 2013 John Wiley and Sons.

Usually, bifunctional catalysts are prepared by impregnating Brønsted acid supports with transition (Co and Ni) or noble metals (Pt and Pd) [116]. The active metal in the catalyst performs the dehydrogenating and hydrogenating functions, and the catalyst carrier has an acidic function. The acidic properties of the catalyst determine its cracking and isomerizing ability. According to the most common mechanism, the resulting *n*-paraffin is first dehydrogenated on the surface of the metal site to *n*-olefins. The molecules diffuse into the Brønsted acid sites, where they are protonated to form carbocations. The carbocation undergoes isomerization or cracking reactions to form an olefin and a carbocation with fewer carbon atoms in the chain. After deprotonation, the olefins are displaced via diffusion to the metal centers, where they are hydrogenated [9,117,118].

The development of effective bifunctional catalysts involves several important aspects. Firstly, the conditions of Fischer–Tropsch synthesis and hydrocracking are significantly different. It is optimal to carry out FTS using cobalt-based catalysts at temperatures lower than hydrocracking. It is necessary to choose the optimal temperature for the two processes simultaneously. Secondly, continuous running of two types of reaction requires close contact between active phase of FT and acidic sites, which is difficult because of catalyst deactivation. For example, a hybrid catalyst consisting of an alkali-promoted molten iron catalyst and HZSM-5 rapidly deactivates due to the migration of the alkali metal from the iron catalyst into the zeolite [119]. Thirdly, the deactivation of the zeolite may be faster due to the deposition of carbon on the acid sites. Thus, improving catalyst stability is a key issue.

To solve the problems of rapid deactivation, a team of authors [120,121] developed a kind of bifunctional catalyst with a core–shell structure. Theoretically, the heavier hydrocarbons formed on the core, which is a conventional FT catalyst, diffuse through the shell, which is a zeolite (Figure 7a,b). Such catalysts have an appropriate acidity to promote the cracking of long chain alkanes and show better stability in the Fischer–Tropsch synthesis. Catalysts with a core–shell structure were also obtained in [122] by depositing a ZSM-5 film of controlled thickness on the Co-$Al_2O_3$ surface. As a result of catalytic tests, an increase in selectivity for motor fuels was observed (the yield of gasoline fraction reached 75.5%), as was a feedstock conversion up to 78.7% due to effective mass and heat transfer, as well as the outer layer of ZSM-5, which is used as a cracking agent and isomerization of hydrocarbons. Core–shell catalysts based on halloysite aluminosilicate nanotubes were also obtained in our work [123]. The formation of bimetallic RuCo nanoparticles inside the support was achieved using microwave radiation. The result was systems with uniform metal deposition and a narrower particle size distribution compared with conventional synthesis methods. In addition, the formation of particles inside the pores of the mineral carrier made it possible to increase the selectivity for liquid fuels (up to 90.0%) and a chain growth index (ASF) of $\alpha = 0.853$, which is typical for the formation of diesel fraction components (Figure 7c).

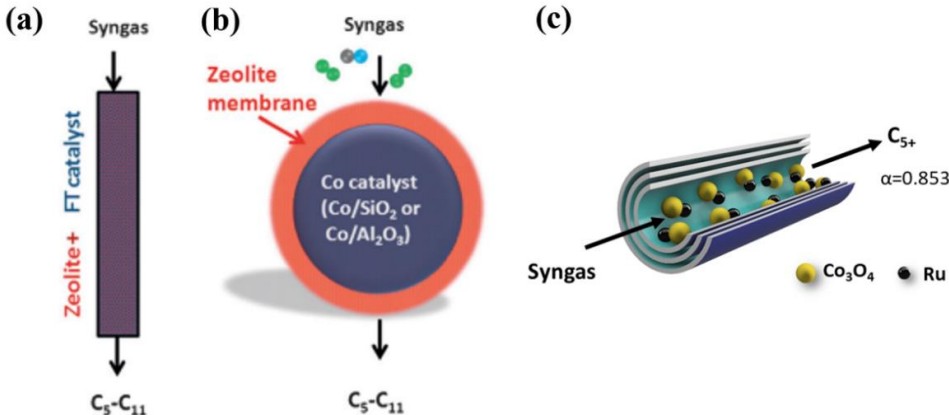

**Figure 7.** Hybrid bifunctional catalyst (**a**) [15], core–shell catalyst (**b**) [15], and core–shell catalyst based on halloysite aluminosilicate nanotubes (**c**) [123]. Reprinted with permission from ref. [15]. Copyright 2013 John Wiley and Sons.

The Co/ZSM-5 catalyst obtained via impregnation based on moisture capacity was encapsulated with a microporous shell of silicalite-1 using hydrothermal synthesis [124]. The synthesized microcapsule catalyst showed a significantly high degree of CO conversion (68.9%) and selectivity for gasoline range hydrocarbons (74.7%) at a low level of methane content due to a homogeneous microporous structure with an additional residence zone for both reactants and product inside its channels. The cobalt-embedded zeolite catalyst described by Liu et al. [125] also showed a high yield on gasoline fuels compared with a conventional Co/SiO$_2$ catalyst and a zeolite supported catalyst due to the limited reaction environment, the high diffusion efficiency, and suitable acidic properties. High selectivity for the diesel fraction (66.2%) was obtained on a Fischer–Tropsch catalyst with cobalt nanocrystals of uniform size embedded in mesoporous SiO$_2$ [56]. The spatial limitation of metal nanoparticles contributes to the inhibition of the aggregation of cobalt nanocrystals during FTS reactions and also leads to an increase in the feedstock conversion. In addition, the contact time between entrapped reaction intermediates and active sites can be increased inside the enclosed space, which further enhances the growth of long chain hydrocarbons. It should be noted that the core–shell catalyst is synthesized mainly using the hydrothermal method, which has some limitations [121]. First, this method is not easy for some zeolites such as HY and Hbeta. Second, the highly alkaline conditions used to synthesize the zeolite and the vulnerability of the FT catalyst core (especially the SiO$_2$ carrier, which can degrade in an alkaline environment) further complicate this method. Thirdly, this method requires large energy and economic costs compared with traditional methods for the synthesis of a Fischer–Tropsch catalyst. Figure 8 shows the selectivity of catalysts for the diesel fraction.

In recent years, many bifunctional zeolite–metal catalysts have also been developed and synthesized for the production of gasoline and diesel fuel. Many studies have found that adjusting the acidity and pore structure of the catalyst results in the selective production of a particular liquid fuel.

Thus, the authors of [15] showed that the higher selectivity for the gasoline fraction in a cobalt catalyst on the carrier H-ZSM-5 is explained by its higher acidity compared with HY and H-modernite. The pore structure of the carrier also plays an important role in the molecular weight distribution of the synthesis products. For example, among cobalt catalysts supported on various zeolites, Co/HZSM-12 showed the greatest efficiency in the synthesis of hydrocarbons in the gasoline range and the Co/HZSM-34 catalyst showed the greatest efficiency in the synthesis of *n*-paraffins, although the first is characterized by lower acidity. This was explained by the difference in the zeolite pore structure: H-ZSM-12 had larger pore channels (0.57–0.61 nm) than H-ZSM-34 (0.50 nm). It was shown that the availability of acid sites located inside the zeolite pores is also significant for the secondary reactions of products.

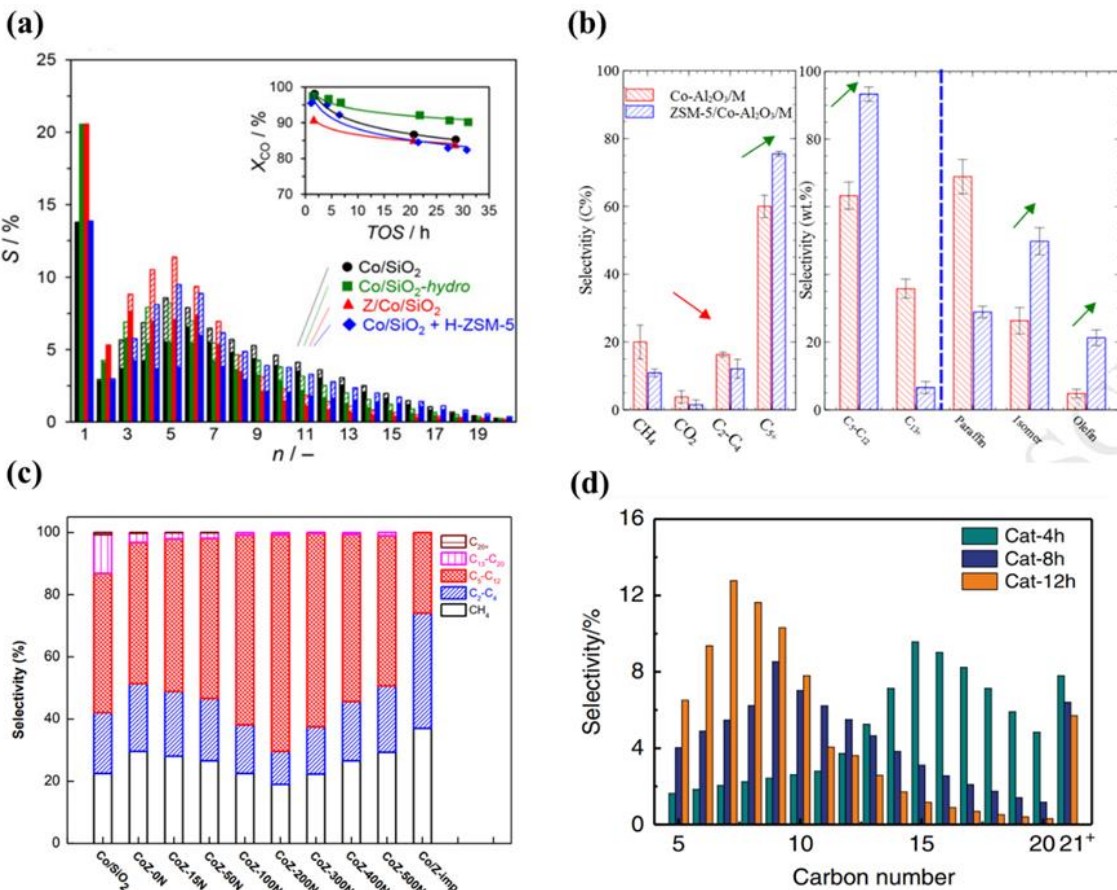

**Figure 8.** Selectivity for diesel fractions of Fischer–Tropsch catalysts published in [121] (**a**), [122] (**b**), [125] (**c**), and [56] (**d**). Reprinted with permission from refs. [121] (copyright 2013 Elsevier), [122] (copyright 2018 Elsevier), [125] (copyright 2016 Elsevier), and [56] (copyright 2018 Springer Nature).

In [126], cobalt-containing catalytic systems were studied in the synthesis of Fischer–Tropsch fuel based on a SAPO-11 microporous molecular sieve with different characteristics of the secondary porous structure. When SAPO-11 is used as a base, CO conversion is at least 76.8% and selectivity for $C_{5+}$ hydrocarbons is more than 59%, so SAPO-11 can be successfully used as an acid component in a bifunctional catalyst. At the same time, the activity of cobalt catalysts based on SAPO-11, as well as the selectivity and stability of the catalytic system, are significantly affected by the characteristics of the secondary porous structure of SAPO-11. Catalyst systems based on SAPO-11 with a more developed secondary pore structure improve the listed process characteristics due to efficient diffusion of the initial reagents and rapid desorption of reaction products. According to the authors, the creation of catalytic systems based on SAPO-11 molecular sieves, which have an even more developed secondary mesoporous structure, opens up the possibility of further increasing the selectivity for the $C_{11}$–$C_{18}$ diesel fraction with good low-temperature properties.

It should be emphasized that diffusion limitations arise in zeolites with micropores, since access to active centers is limited, which leads to restrictions on the activity, selectivity, and service life of the catalyst [127]. To solve this problem, to date, a number of works have been published on the preparation of mesoporous catalysts based on cobalt/zeolite systems for the Fischer–Tropsch process. Comparing microporous zeolites and hierarchical zeolites, it was found that the use of hierarchical zeolites makes it possible to increase CO conversion and selectivity for diesel fractions due to more efficient hydrocracking and isomerization reactions inside mesopores.

It was found that high selectivity for the diesel fraction can be achieved using a cobalt catalyst based on mesoporous zeolite Y in sodium form (Na-*meso*-Y). The selectivity reaches 60%, which is significantly higher than that with the classical ASF distribution (39%). The researchers also revealed that the key factors influencing the FTS are the average size of Co particles and mesopores [98].

Kang et al. [114] compared two methods of depositing cobalt on the carrier: impregnation and melt infiltration. The second method produces cobalt particles with a narrower size distribution. In melt infiltration, cobalt can be selectively deposited in mesopores, which increases the reducibility and dispersion of the particles. The authors suggest that the probability of repeated hydrogenolysis of reaction products can be reduced due to larger pores, which increases selectivity for high-molecular-weight products. The average Na-*meso*-Y mesopore size also affected the product selectivity; with an increase in the average mesopore size, the selectivity for $C_{10}$–$C_{20}$ increased, while the selectivity for methane decreased. The highest selectivity for the diesel fraction is typical for the Co/Na-*meso*-Y catalyst with a pore size of 15 nm prepared via melt infiltration. The authors suggest that the narrower size distribution of Co promotes selective hydrogenolysis and hence $C_{10}$–$C_{20}$ selectivity in the Fischer–Tropsch synthesis.

In one of the studies by Wang et al. [128], the influence of the porous structure of macro- and mesoporous ZSM-5 on the selectivity for gasoline and diesel fractions was studied by using various organic templates in the preparation of the catalyst. The authors emphasized the importance of the number of acid sites, since some acid sites may not perform well during hydrocracking, and an excess of acid sites can lead to excessive hydrocracking in the FTS.

Chen et al. [129] studied a series of Co/MZ hybrid catalysts that differ in the number of acid sites. The systems were obtained by mixing various amounts of nano-H-ZSM-5 zeolite with a Co/MCF catalyst in which nano-H-ZSM-5 zeolite performed the acidic function. As the mass fraction of nano-H-ZSM-5 zeolite increased to 80 wt.%, the number of acid sites increased to $1.96 \cdot 10^{-4}$ mol·g·cat$^{-1}$, which led to a decrease in the formation of $C_{21+}$ products. As the acidity of the catalyst increased, the selectivity for $C_{10}$–$C_{20}$ hydrocarbons decreased due to excessive hydrocracking reactions; the Co/M catalyst without acid sites obeyed the traditional ASF distribution.

In [130], a series of ZSM-5/SBA-15 catalysts with a Co load of 15 wt.% was prepared. The supports were prepared by physically mixing ZSM-5 and SBA-15 in various proportions. The catalytic performance of composite-supported catalysts has been shown to be significantly better than that of the corresponding single-material-supported catalysts. An increase in the content of ZSM-5 from 0 to 20% led to an increase in the selectivity for $C_5$–$C_{22}$ hydrocarbons from 60.5 to 70.0% due to the large pore size of the catalyst, which ensures optimal accessibility to acid sites. A further increase in the proportion of ZSM-5 led to a decrease in liquid products.

In [131], a mesoporous cobalt catalyst based on ZSM-5 with a bimodal structure was synthesized via the double-matrix method. As a result, it was shown that the obtained catalyst with moderate acidity has a higher diesel fraction selectivity and a lower methane yield than the traditional Co/SBA-15 catalyst. Figure 9 shows the selectivity for diesel fractions of Fischer–Tropsch catalysts published in the described articles. Table 6 shows the catalytic characteristics of bifunctional Fischer–Tropsch systems, selective for diesel fractions.

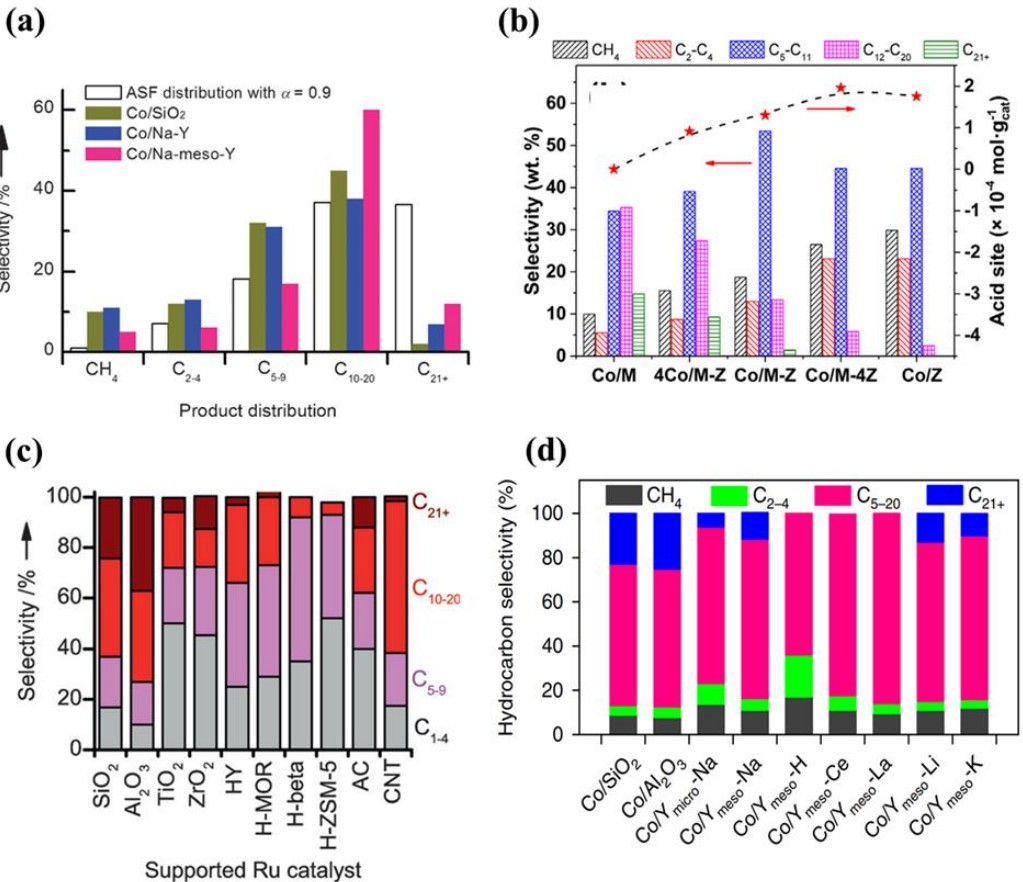

**Figure 9.** Selectivity for diesel fractions of Fischer–Tropsch catalysts published in [98] (**a**), [129] (**b**), [15] (**c**), and [92] (**d**). Reprinted with permission from refs. [98] (copyright 2015 John Wiley and Sons), [129] (copyright 2020 Elsevier), [15] (copyright 2013 John Wiley and Sons), and [92] (copyright 2018 Springer Nature).

**Table 6.** Catalytic characteristics of bifunctional Fischer–Tropsch systems selective for diesel fractions *.

| Catalyst | Loading, wt.% | Reaction Conditions | | | Catalysis Performance | | | Ref. |
|---|---|---|---|---|---|---|---|---|
| | | $T$, °C | $P$, MPa | $H_2/CO$ Ratio | Conversion CO, % | $C_{5+}$ Selectivity, % | $C_{10}-C_{20}$ Selectivity, % | |
| Co-Pt/ZSM-5 | 13Co 0.3Pt | | | | 42 | 82 | 68 | |
| Co-Al₂O₃/ZSM-5 | 13Co | | | | 33 | 52 | 35 | |
| Co-Al₂O₃-Ru/ZSM-5 | 13Co 0.3Ru | 240 | 2 | 2 | 31 | 76 | 55 | [93] |
| Co-Al₂O₃-Pt/ZSM-5 | 13Co 0.3Pt | | | | 41 | 80 | 62 | |
| Co-Al₂O₃-La/ZSM-5 | 13Co 0.3La | | | | 21 | 60 | 46 | |
| Co/SiO₂ | | | | | 65 | 88 | 44 | |
| Co/Al₂O₃ | | | | | 58 | 89 | 41 | |
| Co/HY | | | | | 98 | 75.1 | 38 | |
| Co/Na-Y | 15Co | 230 | 2 | 1 | 80 | 78.2 | 46 | [98] |
| Co/H-*meso*-Y | | | | | 99 | 81.2 | 47 | |
| Co/Na-*meso*-Y | | | | | 94 | 89 | 60 | |
| Co/MZN | | | | | 80.3 | 68.5 | 28 | |
| Co/MZT | 15Co | 220 | 2 | 2 | 75.4 | 68.2 | 35.4 | [128] |
| Co/MZNS | | | | | 70.2 | 41.5 | 10 | |
| Co/MZTE | | | | | 86.2 | 70 | 33.1 | |
| Co/M | | | | | 58.2 | 84.5 | 35.3 | |
| Co/M-Z | 15Co | 250 | 2 | 2 | 25.7 | 68.3 | 13.4 | [129] |
| Co/M-4Z | | | | | 7.5 | 50.4 | 5.9 | |
| Co/Z | | | | | 33.4 | 47.0 | 2.5 | |
| Co/ZS-0 | | | | | 90.0 | 65.1 | 26.6 | |
| Co/ZS-10 | | | | | 90.3 | 68.8 | 29.3 | |
| Co/ZS-20 | 15Co | 240 | 2 | 2 | 90.6 | 79.7 | 38.0 | [130] |
| Co/ZS-30 | | | | | 87.9 | 77.3 | 35.2 | |
| Co/ZS-50 | | | | | 84.9 | 71.9 | 30.7 | |
| Co/ZS-100 | | | | | 70.5 | 69.6 | 34.1 | |
| Co/ZSM-5 | | | | | 28.2 | 45.5 | 10.6 | |
| Co/SBA-15 | | | | | 65.5 | 84 | 42.4 | |
| Co/MZ-1 | 15Co | 220 | 2 | 2 | 30.2 | 74.2 | 31.2 | [131] |
| Co/MZ-2 | | | | | 34.4 | 74.7 | 29.5 | |
| Co/MZ-3 | | | | | 38.2 | 74.1 | 29.1 | |

* Fischer–Tropsch synthesis was performed in a down-flow fixed bed stainless steel reactor.

## 4. Conclusions

The production of diesel fuel through Fischer–Tropsch synthesis is a prospective technology due to the better characteristics of diesel in comparison with petroleum-based products and biodiesel [132]. In this work, we reviewed catalytic systems that make it possible to produce diesel fractions in a one-stage process to eliminate the wax hydrocracking and hydroisomerization step in green diesel production via FTS. The main factors that influence the selectivity of FTS catalysts for selective production of diesel fraction were reviewed. Effective strategies for catalytic systems preparation were discussed.

For Co-based FTS catalysts, Co metal has been proven to be the main active site to promote the FTS chain growth reaction to produce long chain hydrocarbons. At the same time, hcp-Co is characterized by higher activity compared with fcc-Co. Catalysts with particle sizes close to 7–10 nm are the target for optimal Co-based FTS catalysts. On the other hand, systems with a smaller particle size of cobalt (5 nm) in the presence of a hexagonal close-packed crystalline phase provide a higher rate of CO dissociation and selectivity for $C_{5+}$ hydrocarbons compared with fcc-Co. These discrepancies undoubtedly require more fundamental research.

The carrier porosity affects the selectivity of fuels, including the yield of diesel fractions. The microporous structure predominantly leads to the production of gasoline fuels, while the mesoporous material with a pore size of 10–15 nm has a positive effect on mass and heat transfer and promotes the growth of the hydrocarbon chain with the production of diesel fractions. The selectivity of diesel fraction also depends on the acidity of the catalyst, which affects the selective hydrogynolysis of *n*-paraffins. It was found that a moderate amount of medium acid sites leads to a higher yield in relation to the production of diesel fuel.

Promoters are important additives in the preparation of Fischer–Tropsch catalysts. The addition of ruthenium (0.05–0.5 wt.%) reduces the amount of oxygen forms of cobalt, increases the formation of active sites of cobalt ($Co^0$), and promotes easier and faster reduction of cobalt oxide. The addition of Re promotes better hydrogen dissociation and the formation of CH, $CH_2$, and $CH_3$ particles, which leads to an increase in CO conversion. The presence of Ca, Mn, Ce, Zr, and La in the composition of the cobalt catalyst contributes to an increase in the yield of diesel hydrocarbons.

It has been shown that bifunctional catalysts that combine the traditional FT catalyst for hydrogenating CO to heavier hydrocarbons and acid sites capable of catalyzing the hydrocracking of *n*-paraffins have been very promising in recent years for the production of middle distillate liquid fuels. The control of the hydrogenolysis reaction of *n*-paraffins using new solid acid materials with a given porosity and acidity is an attractive method for the selective production of diesel fuel using the Fischer–Tropsch method. When using this method, excellent selectivity for $C_{10}$–$C_{20}$ hydrocarbons ($\approx$60–65%) is achieved. Further research in the development of bifunctional catalysts may lead to an improvement in the selectivity of the diesel fraction under industrial conditions.

**Author Contributions:** Conceptualization, V.S., A.S. and A.G.; writing—original draft preparation, K.M. and A.M.; writing—review and editing, K.M., A.S., A.M., A.G., O.E. and V.S. All authors have read and agreed to the published version of the manuscript.

**Funding:** This research received no external funding.

**Data Availability Statement:** The data presented in this study are available on request from the corresponding author.

**Acknowledgments:** The authors are thankful to the Ministry of Science and Higher Education of RF (FSZE-2022-0002).

**Conflicts of Interest:** The authors declare no conflict of interest.

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
