# Peer review of "Fischer–Tropsch Synthesis Catalysts for Selective Production of Diesel Fraction"

_catalysts, doi:10.3390/catal13081215_

Round 1

Reviewer 1 Report

I want to begin by expressing my appreciation for your effort in this work. The paper explores some interesting ideas and presents a comprehensive analysis of the topic, which is undoubtedly of current interest. However, I have some suggestions to enhance the overall quality and impact of the manuscript.

Positive Aspects:

Firstly, I would like to commend you for the thoroughness and clarity of your writing. Your paper demonstrates a strong understanding of the subject matter, especially evident in the paragraph discussing "The Main Factors Affecting the Selectivity of Diesel Fuels." This section is particularly interesting, comprehensive, and adds value to the paper.

Areas for Improvement:

1.       Introduction: One area that requires improvement is the introduction section. The current version lacks sufficient depth and context to present a clear picture of the state of the art in the field. To strengthen your article's contribution to the sector, it is essential to provide a more comprehensive and engaging introduction. This should outline the existing research landscape, highlight the gaps or limitations in current approaches, and clearly define the objectives of your study.

2.       Conclusions: While your conclusions are reasonable, I believe they would benefit from enrichment, possibly through a comparison with other methodologies for Selective Production of Diesel fraction. By discussing how your approach fares against other existing methods, you can further validate your findings and emphasize the novelty of your research.

3.       Bibliography: I suggest enriching your bibliography with additional relevant and recent references. This will not only strengthen the credibility of your work but also demonstrate a thorough review of the literature related to your topic.

English Language: Although the paper is well written, some aspects of the English language need improvement. I recommend carefully reviewing the manuscript for grammatical errors, sentence structure, and clarity to ensure effective communication of your ideas.

In conclusion, I believe your paper has the potential to make a valuable contribution to the field, especially with the improvements mentioned above. Therefore, I recommend that this paper be accepted after minor revisions. Addressing the points raised will enhance the overall quality, clarity, and impact of your research.

The quality of the English language in the manuscript requires improvement. The language used lacks fluency and appears excessively informal, lacking the scholastic tone expected in academic writing. Furthermore, there are noticeable spelling errors that need to be addressed. It is crucial to undertake a thorough revision of the English language in the paper to ensure its clarity and adherence to academic standards.

Author Response

Authors are thankful to the reviewers for their valuable comments.

Reviewer 1

Areas for Improvement:

  1. Introduction: One area that requires improvement is the introduction section. The current version lacks sufficient depth and context to present a clear picture of the state of the art in the field. To strengthen your article's contribution to the sector, it is essential to provide a more comprehensive and engaging introduction. This should outline the existing research landscape, highlight the gaps or limitations in current approaches, and clearly define the objectives of your study.

 We have improved the introduction section. Pages 1-2.

  1. Conclusions: While your conclusions are reasonable, I believe they would benefit from enrichment, possibly through a comparison with other methodologies for Selective Production of Diesel fraction. By discussing how your approach fares against other existing methods, you can further validate your findings and emphasize the novelty of your research.

 We have modified the conclusion section. Page 22.

  1. Bibliography: I suggest enriching your bibliography with additional relevant and recent references. This will not only strengthen the credibility of your work but also demonstrate a thorough review of the literature related to your topic.

 We have supplemented our bibliography with additional references regarding our topic.

English Language: Although the paper is well written, some aspects of the English language need improvement. I recommend carefully reviewing the manuscript for grammatical errors, sentence structure, and clarity to ensure effective communication of your ideas.

 We have checked the manuscript carefully.

In conclusion, I believe your paper has the potential to make a valuable contribution to the field, especially with the improvements mentioned above. Therefore, I recommend that this paper be accepted after minor revisions. Addressing the points raised will enhance the overall quality, clarity, and impact of your research.

Comments on the Quality of English Language

The quality of the English language in the manuscript requires improvement. The language used lacks fluency and appears excessively informal, lacking the scholastic tone expected in academic writing. Furthermore, there are noticeable spelling errors that need to be addressed. It is crucial to undertake a thorough revision of the English language in the paper to ensure its clarity and adherence to academic standards.

 We have checked the manuscript carefully.

Reviewer 2 Report

This work reviews the Fischer-Tropsch synthesis catalysts for the selective production of diesel fraction. The focus on process conditions, active phase particle size and chemical state, carriers, promoters and the bifunctional catalysts are overviewed. This manuscript is well written, and I recommend it can be accepted after minor revisions. Some comments are listed as follows:

1. The metallic Ru and Co active phase have been reviewed, what about the Fe active phase? Can Fe be included in this review?

2. What about the effect of the interaction between the active phase and the carriers?

3.  It is better to give a few more paragraphs for the part 2.1 Fischer-Tropsch process conditions, which is beneficial for reading.

4.  It is recommended to compare the difference in reaction mechanism between Ru and Co catalyst.

Some English writing errors need to be revised. Such as spelling mistakes and abbreviations for journals.

Author Response

Reviewer 2

This work reviews the Fischer-Tropsch synthesis catalysts for the selective production of diesel fraction. The focus on process conditions, active phase particle size and chemical state, carriers, promoters and the bifunctional catalysts are overviewed. This manuscript is well written, and I recommend it can be accepted after minor revisions. Some comments are listed as follows:

  1. The metallic Ru and Co active phase have been reviewed, what about the Fe active phase? Can Fe be included in this review?

Indeed, for the Fischer-Tropsch process, the most preferred active phases are Fe, Co and Ru. But iron-based catalysts have a high selectivity towards lower olefins, so catalysts of this type are not included. We have added to section 2.2 Active phase particle size and chemical state a clarification on this issue and remark.

  1. What about the effect of the interaction between the active phase and the carriers?

We have completed section 2.3. Carriers with the effects of the interaction of the active phase with carriers and the influence of various carriers on the active phase.

  1. It is better to give a few more paragraphs for the part 2.1 Fischer-Tropsch process conditions, which is beneficial for reading.

We have fulfilled the section 2.1 Fischer-Tropsch process conditions.

  1. It is recommended to compare the difference in reaction mechanism between Ru and Co catalyst.

The detailed description of the reaction mechanism is out of the scope of the work, there are several works that are dedicated to the mechanism of reactions and are reviewed in the manuscript like https://www.mdpi.com/2073-4344/11/3/330.

Comments on the Quality of English Language

Some English writing errors need to be revised. Such as spelling mistakes and abbreviations for journals.

We have checked the manuscript carefully.